# Mild Zika Virus Infection in Mice Without Motor Impairments Induces Working Memory Deficits, Anxiety-like Behaviors, and Dysregulation of Immunity and Synaptic Vesicle Pathways

**DOI:** 10.3390/v17030405

**Published:** 2025-03-12

**Authors:** Jaime Alexander Chivatá-Ávila, Paola Rojas-Estevez, Alejandra M. Muñoz-Suarez, Esthefanny Caro-Morales, Aura Caterine Rengifo, Orlando Torres-Fernández, Jose Manuel Lozano, Diego A. Álvarez-Díaz

**Affiliations:** 1Grupo de Genómica de Microorganismos Emergentes, Dirección de Investigación en Salud Pública, Instituto Nacional de Salud, Bogotá 111321, Colombia; jchivata@ins.gov.co (J.A.C.-Á.); crojas@ins.gov.co (P.R.-E.); 2Grupo de Animales de Laboratorio, Dirección de Producción, Instituto Nacional de Salud, Bogotá 111321, Colombia; ammunoz@ins.gov.co (A.M.M.-S.); ecaro@ins.gov.co (E.C.-M.); 3Grupo de Morfología Celular, Dirección de Investigación en Salud Pública, Instituto Nacional de Salud, Bogotá 111321, Colombia; arengifo@ins.gov.co (A.C.R.); otorresf@ins.gov.co (O.T.-F.); 4Grupo Mimetismo Molecular de los Agentes Infecciosos, Departamento de Farmacia, Facultad de Ciencias, Universidad Nacional de Colombia, Bogotá 11001, Colombia; jmlozanom@unal.edu.co; 5Grupo de Investigación y Desarrollo en Vacunas y Biológicos Estratégicos en Salud Pública, Dirección de Producción, Instituto Nacional de Salud, Bogotá 111321, Colombia

**Keywords:** zika virus, neurodevelopmental disorders (NDDs), behavioral deficits, mild infection model, differential expression, functional enrichment, transcriptomics, Gene Ontology (GO), neurodevelopment

## Abstract

Background: The Zika virus (ZIKV) is an arbovirus linked to “Congenital Zika Syndrome” and a range of neurodevelopmental disorders (NDDs), with microcephaly as the most severe manifestation. Milder NDDs, such as autism spectrum disorders and delays in neuropsychomotor and language development, often go unnoticed in neonates, resulting in long-term social and academic difficulties. Murine models of ZIKV infection can be used to mimic part of the spectrum of motor and cognitive deficits observed in humans. These can be evaluated through behavioral tests, enabling comparison with gene expression profiles and aiding in the characterization of ZIKV-induced NDDs. Objectives: This study aimed to identify genes associated with behavioral changes following a subtle ZIKV infection in juvenile BALB/c mice. Methods: Neonatal mice were subcutaneously inoculated with ZIKV (MH544701.2) on postnatal day 1 (DPN) at a dose of 6.8 × 10^3^ PFU. Viral presence in the cerebellum and cortex was quantified at 10- and 30-days post-infection (DPI) using RT-qPCR. Neurobehavioral deficits were assessed at 30 DPI through T-maze, rotarod, and open field tests. Next-Generation Sequencing (NGS) was performed to identify differentially expressed genes (DEGs), which were analyzed through Gene Ontology (GO) and KEGG enrichment. Gene interaction networks were then constructed to explore gene interactions in the most enriched biological categories. Results: A ZIKV infection model was successfully established, enabling brain infection while allowing survival beyond 30 DPI. The infection induced mild cognitive behavioral changes, though motor and motivational functions remained unaffected. These cognitive changes were linked to the functional repression of synaptic vesicles and alterations in neuronal structure, suggesting potential disruptions in neuronal plasticity. Conclusions: Moderate ZIKV infection with circulating strains from the 2016 epidemic may cause dysregulation of genes related to immune response, alterations in cytoskeletal organization, and modifications in cellular transport mediated by vesicles. Despite viral control, neurocognitive effects persisted, including memory deficits and anxiety-like behaviors, highlighting the long-term neurological consequences of ZIKV infection in models that show no apparent malformations.

## 1. Introduction

Zika virus (ZIKV) is a re-emerging flavivirus transmitted primarily by mosquitoes of the *Aedes* genus, recognized as a significant public health concern [1]. The virus was first isolated in Uganda in 1947, and since then, it has sporadically spread globally. However, it was not until the 2015–2016 outbreak in Latin America and the Caribbean [2] that ZIKV infections were declared a public health emergency of international concern by the World Health Organization (WHO) [3]. To date, 87 countries and territories have reported local transmission of ZIKV. Although infection rates peaked in 2016 [4], ZIKV continues to pose a global health threat, with the potential for further outbreaks in regions where the vector is present, and prior transmission has been reported [3].

ZIKV is primarily an arthropod-borne virus, but it has evolved multiple alternative transmission routes, including vertical (mother-to-child), sexual, bloodborne, and through contact with bodily fluids [5]. The virus has a pronounced tropism for the brain, as established since the early 1950s in animal models [6]. During vertical transmission, ZIKV can infect the developing fetal brain [7], with a preference for neural progenitor cells. In these cells, ZIKV impairs critical neurodevelopmental processes, such as growth, proliferation, and migration, leading to Congenital Zika Syndrome (CZS). CZS is primarily characterized by microcephaly, arthrogryposis, and hypertonia, with more severe outcomes when infection occurs in the first trimester of pregnancy [5,8,9].

CZS occurs in only 5 to 14% of prenatal infection cases [10]. However, the absence of severe structural abnormalities at birth does not ensure healthy development. Normocephalic newborns with normal neuroimaging may later develop neurodevelopmental disorders (NDDs) in early childhood, manifesting as cognitive impairments, learning difficulties, and social behavior abnormalities [11,12,13]. Despite these findings, the molecular and pathophysiological mechanisms underlying subtle behavioral outcomes post-infection remain poorly understood.

In this context, it is crucial to develop infection models that can identify dysregulated genes associated with these mild neurodevelopmental alterations. Studies using immunocompetent mice less than one week old, inoculated through various peripheral routes, have successfully induced neurological symptoms associated with ZIKV infection [14,15,16]. These models have also demonstrated damage in brain regions involved in motor control, learning, and memory, particularly the cortex and cerebellum. A reduction in brain size has been observed during the acute phase of infection with high doses and even with mild doses, which result in offspring without detectable congenital abnormalities, no viral detection in the brain, subtle alterations, and body weights comparable to controls [17,18,19].

Neurodevelopmental events in mice and humans show remarkable age-related correlations. Comparative studies have identified common gene expression profiles at different neurodevelopmental stages in both species [20,21,22]. In rodents, key neurodevelopmental events, such as structural maturation and specialization of neural networks in the prefrontal cortex, as well as the development of maximal gray matter volume and cortical thickness, occur between postnatal days 25 and 30 (DPN). These same processes occur in humans between the ages of 4 and 11 years [23], making the neurodevelopment of a 30 DPN mouse comparable to that of a child in early to mid-childhood.

These molecular approaches, combined with behavioral tests assessing sensorimotor, cognitive, learning, and memory functions, can reveal neurological pathologies. They also provide insight into the molecular and pathogenic characteristics associated with behavioral anomalies in cases of ZIKV infection that lead to normal births but develop into postnatal deficits in neuronal function [18,24]. This approach allows for the characterization of NDDs and the identification of genes that are altered during the infection process.

In this regard, the BALB/c animal model becomes a key biological tool for characterizing neurodevelopmental disorders (NDD) and identifying genes altered during infection, providing crucial information about the disease. Accordingly, the objective of this study is to elucidate the molecular mechanisms underlying these behavioral alterations by integrating gene expression profiles with behavioral assessments in juvenile BALB/c mice, which could contribute to the development of new strategies for its control and prevention.

## 2. Materials and Methods

### 2.1. Ethics Statement and Biosafety

All experimental procedures involving animals in this study were approved by the ethics committee (Minute Number 2, 30 January 2020, Code 35-2019) and the institutional committee for the care and use of laboratory animals (CICUAL, Code R-04-2023) of the National Institute of Health of Colombia (INS). The handling of laboratory animals and tissue samples adhered to the guidelines of the International Guide for the Care and Use of Laboratory Animals [25], and the study followed the principles of the three R’s: Replacement, Reduction, and Refinement [26]. Animal monitoring and welfare were conducted using the format proposed by Van der Meer et al. (2001) [27]. Clinical signs were monitored daily, and euthanasia was performed using a CO2 chamber to minimize animal suffering [28].

All experiments and animal studies involving ZIKV were conducted by qualified personnel in biosafety level 2 laboratories, following standardized biosafety protocols approved by the INS.

### 2.2. Animals and Viral Inoculation

A total of twenty-two BALB/c mice (Charles River Laboratories, Wilmington, MA, USA) were obtained from the animal facility of the National Institute of Health of Colombia (INS) and subcutaneously inoculated with the Zika virus (ZIKV MH544701.1) [14] in a volume of 20 μL on postnatal day 1 (DPN 1); the sex of the animals was not considered a variable. The mice were housed in individually ventilated polysulfone cages with a capacity of 660 cm^3^ (ZyfoneTM, Lab Products Inc., Seaford, DE, USA), enriched with standard mouse bedding (wood shavings and shredded cardboard for nesting material), under controlled environmental conditions of 55% humidity (±5%) and a temperature of 22 °C (±1 °C), with a 12 h light/dark cycle. Water and NIH-31 Open Formula Autoclavable Feed (Zeigler Bros Inc., Gardners, PA, USA) were provided ad libitum.

Initially, a pilot protocol was conducted with 12 BALB/c mice at 1 DPN to refine inoculation techniques, handling, welfare, and behavioral testing, as well as to determine the viral inoculation dose needed to infect the brain and ensure survival beyond DPN 30. Based on previous studies at INS [14,19], which used doses of 6.8 × 10^4^ PFU, experimental doses of 6.8 × 10^3^ PFU and 6.8 × 10^2^ PFU (one hundred and one thousand times lower, respectively) were selected for testing. Control animals were inoculated with a virus-free solution.

Once the inoculation dose was established, an experimental block was designed in which two pairs of adult mice were crossed in separate cages to obtain neonatal mice. Ten individuals were used, with five inoculated with the selected viral dose and the other five mock-inoculated as controls. On day 30 post-inoculation, behavioral tests were conducted, followed by necropsy and collection of the cortex and cerebellum for molecular analysis. All infected animals were from the same litter, as were the control animals. Each mother was housed with her litter in separate cages to ensure isolation between the control and infected groups.

### 2.3. Viral Replication and Titration

VERO E6 cells (ATCC CCL-81, Manassas, VA, USA) were cultured in Minimum Essential Medium (Gibco, Grand Island, NY, USA), supplemented with 1% HEPES (Gibco, Bleiswijk, The Netherlands), sodium bicarbonate, and 2% fetal bovine serum (Gibco, Grand Island, NY, USA). These cells were then infected with the ZIKV strain (Zika_virus_459148_Meta_Colombia_2016, GenBank: MH544701.1), which was isolated from a pregnant patient diagnosed with ZIKV during the Colombian epidemic [14]. The strain was supplied by the Cellular Morphology group at INS (obtained by Colciencias Code: 210474455818, INS-Colciencias-Salutia Contract No. 672 of 2017). Five days post-infection, the supernatants were collected, centrifuged at 300× *g* for 5 min, and filtered through a 0.2 μm membrane to remove cellular debris. The supernatants were then cryopreserved at −80 °C until further use. Plaque assays were performed with seven serial dilutions (10-fold) to determine viral titers, expressed as plaque-forming units (PFU).

### 2.4. Necropsy and Sample Preservation

A longitudinal incision was made from the occipital region of the skull to the nasal cavities. Cuts were then made on both sides of the skull to separate the hemispheres. Once exposed, the brain was carefully lifted from its base, severing the cranial nerves holding it in place [29]. To separate the cerebellum, cuts were made at the most caudal part of the cerebral hemispheres, isolating the cerebellum from the brainstem as described by Hamel and Cvetanovic (2020) [30]. The dissection of the cortex followed the protocol proposed by Beaudoin et al. (2012) [31]. The tissues were preserved in TRIzol Reagent (Invitrogen Life Technologies, Carlsbad, CA, USA) and stored at −80 °C for future use.

### 2.5. Detection of ZIKV by RT-qPCR and Genome Quantification

Total RNA from the cortex and cerebellum was extracted using the RNeasy Lipid Tissue Mini Kit (Qiagen Benelux, Venlo, The Netherlands), following the manufacturer’s instructions. Quantification was carried out using a Qubit^®^ fluorometer (Invitrogen, Waltham, MA, USA) and a Nanodrop (Thermo Scientific, Waltham, MA, USA). Molecular detection of ZIKV in the cerebellum and cortex was performed using one-step RT-qPCR, as described by Álvarez-Díaz et al. (2021) [32]. The final reaction volume of 20 μL contained 10 μL Luna Universal Probe One-Step (NEB, Ipswich, MA, USA) Reaction Mix (2X), 1 μL Luna WarmStart RT Enzyme Mix (20X), 2.5 μL Probe-Primer mix (ZIKV), 5 μL of nuclease-free water, and 1.5 μL of RNA. The thermal profile was set at 55 °C for 10 min, 95 °C for 2 min, followed by 40 cycles of 95 °C for 15 s and 55 °C for 45 s for extension. All samples were evaluated in triplicate.

Viral genome quantification was determined using a standard curve with a 10-fold dilution factor, generated using a DNA construct designed by Macrogen (Seoul, Republic of Korea) [32]. Calculations followed the Applied Biosystems protocol [33], interpolating Ct values obtained from the viral detection PCR using known concentrations of the DNA construct. The final viral genome counts were expressed as the number of viral copies per nanogram of total RNA.

Relative quantification was also performed to normalize the viral copy number against the expression of the constitutive gene GAPDH. The thermal profile for this reaction was the same as previously described, with the addition of a melt curve from 65 °C to 95 °C, increasing by 0.5 °C every 0.05 s, due to the use of the SYBR Green detection method. The sequences of the primers and probes used are listed in Table 1.

### 2.6. Library Preparation and Sequencing

Library construction was carried out automatically using MGI technology (MGISP-100 system, MGI. Shenzhen, China) in accordance with the manufacturer’s instructions. High-quality RNA, with a RIN value ≥ 7 and optimal purity (OD 260/280 ratio between 1.8 and 2.0 and OD 260/230 ratio ≥ 2 for phenolic contaminants), was used for ribosomal RNA depletion. This process involved target RNA hybridization to a probe and RNase H digestion of rRNA, followed by DNA digestion with DNase 1 and target RNA purification using magnetic beads. Subsequently, the purified RNA was fragmented, reverse-transcribed, and the second strand synthesized and purified. Poly-A tails were repaired, adapters were ligated, and the ligated products were amplified via PCR. PCR products were then purified with magnetic beads, and the libraries were quality-controlled before proceeding with single-stranded fragment circularization.

High-throughput sequencing was performed using the DNBSEQ-G50RS (MGI, Shenzhen, China) system with an SE 100 flowcell, spanning 120 cycles. DNA nanoballs (DNBs) were synthesized from the single-stranded DNA circularization obtained during library preparation to initiate sequencing. Library concentration was maintained at ≥2 fmol/μL, with each DNB reaction requiring 40 fmol of library in a 100 μL volume. Sequencing required a minimum DNB concentration of 12 ng/μL.

### 2.7. Differential Expression and Protein–Protein Interaction Network Analysis

Quality control of the sequencing data was performed using Trim_Galore [34], retaining all reads with a quality score (Q) > 30. Filtered reads were aligned to the *Mus musculus* reference genome (GRCm39) using Bowtie2 [35]. The expression matrix was generated with FeatureCounts [36], and differentially expressed genes (DEGs) were selected based on log2 (fold change) > 2 or log2 (fold change) < −2, with statistical significance (*p*-value < 0.05) determined using DESeq2 version 1.40.2 [37]. To assess the biological functions of the identified DEGs, WebGestalt 2024 (WEB-based Gene SeT AnaLysis Toolkit) [38] was used to identify enriched Gene Ontology (GO) biological processes.

To explore molecular interactions among the DEGs, networks were constructed separately for upregulated genes in both the cortex and cerebellum using STRING version 12.0 [39]. Key parameters included a confidence score threshold of 0.7 and interaction sources limited to Experiments, Databases, and Co-expression. Nodes were grouped using MCL clustering. Further network refinement and analysis were conducted using Cytoscape version 3.10 [40]. We applied Degree, Betweenness, and Closeness Centrality measures to identify central hub genes. Additionally, to identify common patterns, we merged and compared networks of upregulated DEGs in both the cortex and cerebellum.

### 2.8. Behavioral Testing

All behavioral tests were conducted during daytime hours, between 9:00 AM and 3:00 PM, in the procedure room of the INS animal facility, under artificial white light. Only the animal handling technician was present during the procedures. The animals were transferred to the experimental room 5 to 30 min before testing began. Devices were cleaned with 50% ethanol between animals to eliminate olfactory traces. The tests were recorded using a Monochrome Industrial DMKAUX287 camera (The Imaging Source, Bremen, Alemania), and the videos were processed using Any-Maze software version 7.35 (Stoelting Co., Wood Dale, IL, USA).

#### 2.8.1. T-Maze (Ugo Basile, Gemonio, Italy)

The spontaneous alternation methodology was employed to assess working memory, defined as the short-term storage of information essential for completing specific tasks. This method does not require rewards, punishments, or prior training. Instead, it relies on the natural tendency of rodents to explore novel arms rather than familiar ones, prompting them to alternate their arm choices in consecutive trials. This behavior demonstrates their ability to recall previously visited arms [41,42]. The methodology is highly sensitive to a broad range of brain injuries, as exploring new environmental stimuli requires the functionality of both limbic and non-limbic pathways, including the prefrontal cortex and cerebellum [43].

The apparatus consisted of a T-shaped maze with a start arm and two lateral goal arms. Mice were released at the stem of the maze (start arm) and allowed to explore one of the two goal arms. Upon entering a chosen arm, a guillotine door was closed to confine the animal for 30 s. The mouse was then returned to the start arm, with all doors raised. Trials were conducted with an intertrial interval (ITI) of 50 s. Each experiment included an initial trial (T0) followed by six repetitions [42].

#### 2.8.2. Rotarod Test (Ugo Basile, Gemonio, Italy)

We used the continuous acceleration protocol to assess coordination, muscle strength, and motor control in relation to cerebellar integrity [44]. In this test, mice must maintain their balance on a rotating rod that accelerates continuously. The rod has a diameter of 3 cm, is suspended 30 cm above the base, and features parallel ridges to allow animals to grip effectively [45]. This method is widely used to assess ataxia, a condition characterized by impaired motor coordination caused by significant cerebellar damage [46].

The device was initially set to a speed of 4 rpm, with an acceleration rate of 20 rpm/min. Each mouse was placed on the rod, held by the tail, and positioned to face away from the direction of rotation to encourage forward walking. The system began accelerating 10 s after positioning the mouse and recorded the latency to fall. If a mouse stopped walking forward and instead clung to the rod, completing passive rotations, the latency was recorded as the time when the mouse completed one full passive rotation [45,47].

#### 2.8.3. Open Field Test

This test primarily assesses locomotor behavior, muscle strength, and anxiety. It is one of the most widely used behavioral tests in animal studies due to its simplicity, quick setup, and ability to generate diverse behavioral data. The apparatus consists of a square area measuring 30 × 30 cm, enclosed by walls high enough to prevent escape [48,49].

The open field test requires no prior training and enables the evaluation of various behavioral parameters. Locomotion is the most frequently studied aspect in the open field test. Mice were allowed to move freely within the apparatus, and their movements were tracked throughout the trial. The protocol involved a 5 min observation period, during which uninterrupted free movement of the animal was recorded on video. The mouse was initially placed in the center of the field, with a 5 s acclimation period before the technician left the room to ensure the tracking software detected only the animal [48].

### 2.9. Statistical Analysis

Statistical analyses for behavioral tests and viral copy number quantifications were performed using GraphPad Prism version 6.0 (GraphPad Software, La Jolla, CA, USA). All comparisons between groups were conducted using the Mann–Whitney U test. A significance level of *p* < 0.05 was adopted for all tests. Data were graphically represented as mean ± SEM. For differentially expressed genes between simulated and infected samples, a false discovery rate (FDR) ≤ 0.05 was used as the cutoff criterion to avoid false positive results.

## 3. Results

### 3.1. Subcutaneous Inoculation with ZIKV in Neonatal Mice Induces Brain Tissue Infection Without Fatal Outcomes

At 10 days post-inoculation (DPI 10), RT-qPCR revealed the presence of viral RNA in both inoculation doses (6.8 × 10^2^ and 6.8 × 10^3^ PFU). However, earlier cycle thresholds (CT) were observed with the higher dose for both the cortex and cerebellum. The number of viral genomes was determined using a standard curve, normalized against the constitutive gene GAPDH for each dose. Although no significant differences were found between the quantified genome numbers (Figure 1A), the means were higher for the higher dose. Additionally, the lower dose exhibited more dispersed data and a higher coefficient of variation, with 114.2% in the cerebellum and 81.73% in the cortex, compared to 42.64% in the cerebellum and 29.4% in the cortex for the 6.8 × 10^3^ PFU dose.

Animal welfare monitoring indicated that both doses allowed survival beyond day 30. From DPI 20 onwards, abnormal clinical and phenotypic signs were observed, such as slightly closed eyes, slow response to contact, reduced fear response to handling, mild gait abnormalities, and slight body tremors. Considering the molecular results, the 6.8 × 10^3^ PFU dose was selected for further investigation. This dose not only allowed survival beyond DPI 30 but also exhibited more homogeneous viral genome counts with a lower coefficient of variation among individuals.

### 3.2. Infection Model with Heterogeneous Clinical Response and Decrease in Viral Genomes over Time

Welfare evaluations revealed that ZIKV-infected animals exhibited less weight gain compared to control animals (Appendix A). Infected animals displayed clinical signs like those observed in the previous experiment, such as slightly closed eyes, slow response to contact, reduced fear response to handling, mild gait abnormalities, and slight body tremors. Notably, the group of infected animals showed individuals with varying degrees of clinical signs, ranging from no noticeable symptoms to mild manifestations of the disease. Importantly, all infected animals were from the same litter, meaning they were the offspring of the same mother, housed in the same cage, and inoculated on the same day with identical conditions of volume (20 μL) and dose (6.8 × 10^3^ PFU). However, since our objective was to develop a model of subtle manifestations, the five animals with the least clinical symptoms, which appeared healthy, were selected for behavioral and molecular testing.

RT-qPCR for determining the number of ZIKV viral copies showed an efficiency of 95.6% in the absolute quantification curve. Statistically significant differences in viral genome numbers were observed between animals evaluated at DPI 10 and DPI 30 for both the cortex and cerebellum. The mean viral copies per nanogram of total RNA were higher in animals evaluated at DPI 10 compared to those at DPI 30. Specifically, there were 8.4 × 10^6^ copies for the cortex at DPI 10 versus 1.2 × 10^4^ copies at DPI 30 and 4.4 × 10^6^ copies for the cerebellum at DPI 10 versus 1.5 × 10^4^ copies at DPI 30 (Figure 1B). These results indicate a reduction in viral copies in both brain tissues over time, suggesting a potential immunological resolution of the infection.

### 3.3. Behavioral Abnormalities in ZIKV-Infected Mice: Altered Cognitive but Not Locomotor Responses

In the T-maze test (Figure 2A), which evaluates short-term memory, the percentage of alternations was significantly higher in the mock DPI group compared to the infected animals, suggesting a cognitive anomaly in the latter. The mean alternation for the mock group was 70.2%, while it was 37.5% for the infected animals. The rotarod test (Figure 2B) showed no statistical significance; however, there was a trend toward shorter fall latencies in the infected (ZIKV) group compared to the control. The mock group had a mean latency of 115 s, whereas the infected group averaged 78 s. In the open field test (Figure 2C), the infected animals showed a preference for remaining in the peripheral zone of the apparatus. There was a significant difference between the groups, with the ZIKV group averaging 5.8 m compared to 7.5 m for the mock group. On the other hand, the results from Figure 2D,E suggest no motor impairments in the infected animals, as no significant differences were observed between the groups in total distance traveled (Figure 2D) or average speed during the test (Figure 2E).

### 3.4. Overexpression of Genes Associated with Immune Response and Downregulation of Genes Associated with Vesicular Functions and Cellular Architecture

From cDNA libraries generated through high-throughput sequencing of cortex and cerebellum tissues, the differentially expressed genes (DEG) were identified considering DEG those with a *p*-value of <0.05 and a log2(fold change) > 2 or <−2 using DESeq2 (version 3.7). This approach allowed us to explore alterations in the brain transcriptome of our ZIKV-infected mouse model, which exhibited an apparently healthy phenotype compared to mock-inoculated controls. A total of 309 differentially expressed genes (DEGs) were identified. In the cortex, 120 DEGs were detected, with 91% upregulated and 9% downregulated. In the cerebellum, 189 DEGs were found, with 75% upregulated and 25% downregulated (Appendix A).

Gene Ontology (GO) enrichment analysis revealed that Zika virus infection activates processes related to cytoskeletal organization, intracellular transport, and immune response. Upregulated genes were involved in pathogen–host interactions, including immune responses and cytoskeletal organization, while downregulated genes were associated with cellular metabolism, protein synthesis, and molecular functions linked to ribonucleoprotein assembly and translation (Appendix A).

To determine the functional enrichment of these DEGs, we conducted Gene Ontology (GO) enrichment analysis. In the cortex, functions enriched in the upregulated DEGs within the Biological Processes category included immune responses, such as response to interferon-beta and interferon alpha, antigen processing and presentation, defense response to virus and protozoa, and cellular response to cytokine stimulus. These findings underscore the activation of the innate immune system in this region as a response to infection. Conversely, in the downregulated DEGs, enrichment was observed in processes such as regulation of chromatin organization, chemokine-mediated signaling, and cell migration, which suggest a suppression of cellular remodeling and signaling pathways potentially critical for immune cell mobilization. The Molecular Function category in downregulated genes showed significant enrichment in chemokine activity and G-protein-coupled receptor binding, reflecting the disruption of intercellular communication mediated by chemokines (Figure 3A). In the cerebellum, the upregulated DEGs were enriched in biological processes like those observed in the cortex, with emphasis on response to interferon-beta, antigen processing and presentation, and innate immune response. However, there was an additional enrichment in broader processes, such as regulation of responses to biotic stimuli, highlighting the cerebellum’s activation of immune pathways in response to viral infection. Downregulated DEGs in the Cellular Components category showed enrichment in processes associated with intracellular transport, such as vacuolar membrane, coated vesicle membrane, and COPI vesicle coat (Figure 3B). This suggests that ZIKV infection may interfere with vesicle trafficking pathways in the cerebellum, potentially aiding viral replication and escape from host defenses. Comparatively, both tissues displayed enrichment in immune-related biological processes for upregulated genes, such as response to interferons and defense against pathogens, indicating a common antiviral immune activation mechanism. However, differences were evident in the downregulated processes: while the cortex showed suppression of chromatin organization and chemokine signaling, the cerebellum displayed disruption in vesicle-mediated transport.

### 3.5. Interaction Network of the Top DEGs Expressed in the Cortex and Cerebellum

To investigate the interactions of the DEGs identified in the cortex and cerebellum, we constructed individual networks for each tissue and a merged network to explore global interactions. Using STRING, a high-confidence interaction score threshold of >0.7 was applied to ensure robust functional relationships, while unconnected nodes were retained for completeness. The MCL clustering algorithm grouped nodes into distinct functional and interaction modules. Within these networks, we identified the top 10 hub genes with higher degree and betweenness centrality. Table 2 and Table 3 list the names and functions of these hub genes upregulated and downregulated, respectively. Differences in gene interaction organization were observed between the two tissues, revealing unique and shared immune mechanisms.

To comprehensively analyze the interactions of DEGs in both the cortex and cerebellum, we constructed a merged network, integrating findings across tissues to provide a unified view of the brain’s immune and neuronal response to ZIKV infection. The merged network revealed four distinct clusters, with most upregulated genes grouped into modules representing antiviral pathways and antigen presentation processes. This highlights the overlap and coordination of immune and antiviral responses across both tissues (Figure 4B). The gene distribution indicated that 64.17% were unique to the cerebellum, 4.17% were specific to the cortex, and 31.67% were core genes shared between both tissues (Figure 4A).

Core genes were mainly localized within the cluster associated with interferon-beta response and antiviral defense mechanisms. Notably, hub genes such as *Ifit3*, *Stat1*, *Ifit1*, and *Usp18* played central roles in coordinating immune responses across the brain, regulating antiviral signaling pathways, particularly those driven by interferon responses.

The cerebellum exhibited additional clusters related to antigen presentation pathways, including genes like *Ctss* (Cathepsin S), *B2m* (Beta-2 microglobulin), and *H2-Q* family members, which are critical for major histocompatibility complex (MHC) and T cell activation. Genes such as *C3* and *LY86* indicated a focus on complement activation and cellular immune responses, reflecting the cerebellum’s specialized immune role. In contrast, the cortex-specific module, though smaller, contained genes like Gbp5, Oas2, and *Trim25*, which are key regulators of antiviral responses and inflammation. For example, *Trim25* plays a crucial role in activating RIG-I signaling, a vital pathway for detecting viral RNA and initiating innate immune defenses. This suggests the cortex may have a greater emphasis on innate immune activation and inflammatory modulation.

The individual tissue-specific networks provided additional insights into the organization of gene interactions within each tissue (Appendix A). The cortex network displayed four clusters, primarily involving antiviral responses and immune activation. Genes such as *Ifit3*, *Ifit1*, *Ifi44*, *Rtp4*, and *Rsad2* formed a central hub in the red cluster, underscoring their importance in interferon-mediated antiviral mechanisms. Similarly, the cerebellum network, with five clusters, exhibited greater specialization in antigen presentation and cellular trafficking. Hub genes like *Ifih1*, *Ifi44*, and *Usp18* played central roles in the cerebellum’s antiviral response, with an additional focus on antigen presentation pathways involving Ctss and H2-Q family members.

### 3.6. Interaction Networks of Downregulated DEGs in the Cortex and Cerebellum

To explore the downregulated DEGs across both brain regions, we applied the same criteria and methods used for the upregulated analysis. The merged network revealed five major clusters integrating neuronal function, immune regulation, and intracellular trafficking processes (Figure 5). These clusters emphasize how ZIKV disrupts neuronal and immune mechanisms across both brain regions. The distribution of downregulated genes showed that 80% were specific to the cerebellum, 7.27% to the cortex, and 12.73% were core genes shared between both tissues.

*Xist*, *Phpt1*, *Peak1*, and *Tsix* emerged as core genes, reflecting their involvement in distinct functional modules (Appendix A). For example, *Xist* and *Tsix* were located in the “Phosphotyrosine-binding domain” cluster, indicating their potential relevance to transcriptional regulation through phosphotyrosine signaling. *Phpt1* was associated with protein phosphorylation and intracellular signaling, located near genes such as *Elk1* and *Prkcg*, both involved in signal transduction.

In the cerebellum, clusters were related to motor coordination and intracellular trafficking, as evidenced by genes like *Atp6ap1*, *Atp6v1g2*, *Tmem199*, *Rnasek*, *Dao*, and *L3hypdh* (blue cluster in Figure 5). These genes are primarily involved in intracellular transport, acidification, and motor coordination. Their downregulation suggests impaired endosomal function, potentially facilitating ZIKV replication.

In the cortex, downregulated genes such as *Shank1*, *Ccl26*, *Cxcl14,* and *Ppbp* were identified. These genes play key roles in synaptic plasticity, immune signaling, and neuronal communication. The significant downregulation of *Shank1* may indicate impaired synaptic stability and plasticity, which are essential for learning, memory, and cognitive development. Meanwhile, the downregulation of *Ccl26* and *Cxcl14*, which are involved in immune signaling, suggests a potential ZIKV-induced mechanism of immune evasion.

## 4. Discussion

Zika virus (ZIKV) is widely recognized as a neurotropic virus capable of inducing neurological defects, both in human and animal models. However, the molecular mechanisms underlying the subtle neurodevelopmental effects in individuals born without severe malformations but later diagnosed with neurodevelopmental disorders remain poorly understood. Our study presents an experimental model that simulates the mild clinical manifestations observed in some human cases of congenital ZIKV infection, providing a platform to investigate the molecular basis of these significant yet subtle impacts on brain function, as well as a potential tool to identify targets to consider in the development of treatments aimed at controlling and preventing this disease.

It has been reported that intraperitoneal inoculation of BALB/c mice with a strain of ZIKV from the 2016 epidemic in Colombia induced severe clinical manifestations, including significant weight loss, tactile hypersensitivity, action tremors, and gait instability by the seventh day post-infection (dpi), culminating in severe paralysis and mortality by dpi 11 [14,19]. In the present study, a ZIKV infection model with moderate symptoms was established by modifying the route of inoculation from the intraperitoneal to the subcutaneous pathway and using non-lethal doses lower than those previously reported by our group [14,19]. In this new model, the animals did not exhibit weight loss (Appendix A), and clinical signs appeared only up to 20 dpi, presenting mild to moderate characteristics. Furthermore, a significant reduction in ZIKV genomes was observed in the cortex and cerebellum by dpi 30 compared to dpi 10 (Figure 1B), indicating an immunological resolution of the infection.

In line with this, our model demonstrates that subcutaneous inoculation of neonatal mice with sublethal doses of ZIKV leads to brain infection without fatal outcomes. This survival allowed us to observe behavioral and molecular changes over time, revealing a correlation between viral persistence, immune response, and neurocognitive defects. While previous studies have predominantly focused on severe outcomes like microcephaly, our model provides novel insights into how ZIKV infection can cause cognitive impairments without visible clinical manifestations.

It has been determined in other mouse strains [50,51] that inoculation with moderate doses of ZIKV on postnatal day 1 can induce a transient neurological syndrome. The affected animals exhibited unsteady gait, kinetic tremors, severe ataxia, and seizures between 10 and 15 days post-infection; however, these symptoms resolved approximately one week later. These studies also reported reduced weight gain, motor function impairments assessed through open field and rotarod tests, and hyperactivity-associated behaviors, along with the expression of genes linked to interferon response, antigen presentation via MHC, and inflammation. In contrast, in our study, the animals did not exhibit distinct symptomatic stages or weight loss (Appendix A). Instead of hyperactivity-related behaviors, our models displayed anxiety-associated behaviors (Figure 2). Furthermore, similar biological categories related to immune response were identified compared with our study.

At the molecular level, transcriptomic analysis revealed significant alterations in gene expression, particularly those related to immune responses, cytoskeletal organization, and vesicle-mediated transport. Upregulation of interferon-stimulated genes (ISGs) and immune-related pathways in both the cortex and cerebellum indicates an active antiviral defense mechanism in response to ZIKV infection (Figure 3). Indeed, it was suggested that the top 10 upregulated genes in ZIKV-infected mice encode proteins involved in the interferon (IFN)-mediated antiviral response, playing essential roles in viral recognition, immune signaling, and inhibition of viral replication. Ifit1, Ifit2, Ifit3, Rsad2, Oasl2, and Ifih1 function as antiviral effectors, detecting viral RNA and inhibiting viral replication. Stat1 and Irf7 are key transcriptional regulators of IFN signaling, driving the expression of interferon-stimulated genes (ISGs). Usp18 regulates ISG15-conjugation homeostasis, while Ifi44 contributes to cytoskeletal organization [52,53,54].

However, immune activation during ZIKV infection has been reported by other authors [55,56,57,58,59] and appears to be associated with the downregulation of genes related to cellular metabolism, potentially compromising neuronal function. It is believed that the release of soluble immune mediators physiologically modulates the activity of neuronal networks, influencing learning and memory processes by regulating synaptic transmission and long-term plasticity [60]. This hypothesis can also be supported by the fact that the effective functioning of neuronal and immune pathways requires an energy supply [61]. Neurons, among all cell types in the body, are the highest consumers, given that 80% of the glucose that enters the brain is used in the glutamine–glutamate cycle, the main neurotransmitter mediating neuronal excitation [62,63]. In humans, the same finding has been confirmed, as under pentobarbital anesthesia, glutamate release is inhibited, leading to a 50–70% decrease in glucose utilization [63]. In this same regard, some authors consider the nervous and immune systems to be ’selfish systems’ when it comes to sharing energy resources to overcome external threats [61].

One of the most notable findings in our study is the differential impact of ZIKV infection on the cortex and cerebellum. Although both regions displayed immune activation, the cerebellum exhibited greater involvement in antigen presentation and complement activation pathways (Figure 4), suggesting a specialized role in coordinating immune responses during ZIKV infection. Research using Theiler’s murine encephalomyelitis virus demonstrated that the immune response mediated by MHC I provides greater protection in brain regions rich in white matter or glial cells, such as the corpus callosum, brainstem, and cerebellum [64]. In this regard, the exacerbated immune activation response found in the cerebellum in our study could also contribute to protecting the brain from the drastic changes associated with ZIKV, while at the same time being linked to the cognitive deficits observed in infected animals, considering the previously stated interplay between the immune system and the nervous system [60,61].

The findings of our study (Figure 4) also allow us to observe a group of genes associated with antigen presentation and processing, a process that can lead to the activation of T lymphocytes in the brain [65]. This activation triggers an inflammatory cascade mediated by pro-inflammatory cytokines, such as IFN-γ, IL-6, and TNF-α [66,67]. The inflammation resulting from immune activation can interfere with synaptic plasticity, a critical process for memory and learning [68,69].

In contrast, the cortex-specific gene expression highlighted a stronger emphasis on antiviral responses and inflammation, indicating a more direct role in controlling viral replication and modulating immune responses. However, it is not well understood whether the clinical alterations caused by ZIKV, frequently found in certain brain areas like the cortex [70,71,72], are exclusively associated with the virus’s tropism or if they are also linked to the host’s immune response profile in each brain area [73,74].

On the other hand, the repressed biological functions associated with vesicles in the cerebellum reflect marked synaptic dysregulation (Figure 3). It is well known that V-ATPases play a crucial role in neurotransmission by generating a proton gradient across the synaptic vesicle membrane, a process essential for neurotransmitter loading and fusion of loaded vesicles [75,76]. Additionally, clathrin-coated vesicle membranes are fundamental for recycling synaptic vesicles, enabling their retrieval after exocytosis and ensuring a constant supply of vesicles for neurotransmission [77,78,79].

Hence, the observed phenotypic results could be related to the repressed biological categories identified in the functional enrichment analysis, both combined (Figure 5) and individually for the cortex and cerebellum (Figure 3). In the cerebellum, seven categories linked to synaptic membrane and vesicle formation and function were identified: “Proton-Transporting V-Type ATPase Complex”, “Vacuolar proton-transporting V-type ATPase complex”, “Coated vesicle membrane”, “COPI vesicle coat”, “Vacuolar membrane”, “Cytoplasmic vesicle membrane”, “Clathrin-coated vesicle membrane”, and “Postsynaptic membrane”. Additionally, a category associated with cellular architecture, “Anchoring junction”, was identified. In contrast, the cortex showed categories related to functional and structural processes, such as “Regulation of chromatin organization”, “Chemokine-mediated signaling pathway”, “Cell migration”, and “Protein-coupled receptor binding”.

Also striking in our results is the suppression of the Shank1 gene in the cortex, which has been associated, along with other members of its family, with several key functions in neurological development, including structure, signaling, and synapse formation, as it is part of the proteins that constitute the postsynaptic scaffolding networks [80]. In a Shank1 knockout mouse model used to study the autism spectrum, mild anxiety-like behaviors were identified in 6- to 7-week-old individuals, which is consistent with our findings. However, unlike our results, that study reported alterations in motor function [81].

Our behavioral tests support the idea that ZIKV infection leads to cognitive impairments, particularly in tasks involving working memory and spatial exploration. The significant reduction in alternation percentage in the T-maze test, along with the increased preference for peripheral exploration in the open field test, indicates that ZIKV-infected animals exhibit cognitive inflexibility and anxiety-like behaviors. These findings are consistent with clinical reports of neurodevelopmental disorders in children exposed to ZIKV in utero, who often display learning difficulties, attention deficits, and heightened anxiety [82]. Despite the broad immunological response observed in the cortex and cerebellum and the absence of clinical signs, behavioral test results revealed disparities between control and infected animals. It has been reported that neurological problems during neurodevelopment caused by ZIKV leave latent sequelae in adulthood, even in animals with complete resolution of the infection [18]. In the T-maze test (Figure 2A), infected animals showed a higher percentage of working memory errors by choosing the unvisited arm. In the open field test (Figure 2C), infected animals traveled a greater distance in the peripheral area of the maze, a behavior associated with anxiety, primarily mediated by the cortex [49,83,84,85,86].

It is important to highlight that this study focused on the bioinformatic identification of a broad set of differentially expressed genes as a consequence of mild ZIKV infection and their possible association with behavioral abnormalities. One of the main limitations of this work is the lack of protein-level validation of the identified genes, as it is well known that not all transcripts are translated into proteins due to various transcriptional and post-transcriptional regulatory mechanisms. Additionally, while these genes may be related to the observed behavioral alterations, further studies are needed to confirm their causal role, considering that these behavioral changes may also be influenced by immunological factors such as neuroinflammation. Nevertheless, this study provides valuable insights by identifying potential molecular targets that can be explored in future research to better understand the long-term effects of ZIKV infection in individuals with normocephalic births.

## 5. Conclusions

Our study demonstrates that mild ZIKV infection in neonatal mice can induce working memory deficits and anxiety-like behaviors without affecting motor function. At the molecular level, we identified a downregulation of genes related to synaptic vesicle activity, suggesting a potential disruption in neuronal communication and synaptic plasticity. Additionally, we observed strong immune activation in the cortex and cerebellum, with an inflammatory response that could contribute to the detected cognitive impairments. Unlike previous studies reporting severe motor manifestations following ZIKV infection, our findings indicate that even mild infections can generate subtle cognitive impairments related to memory and anxiety behaviors. These results highlight the importance of further investigating the underlying mechanisms of ZIKV infection and its impact on neurodevelopment, as well as the need to validate these findings at the functional and protein levels.

## Figures and Tables

**Figure 1 viruses-17-00405-f001:**
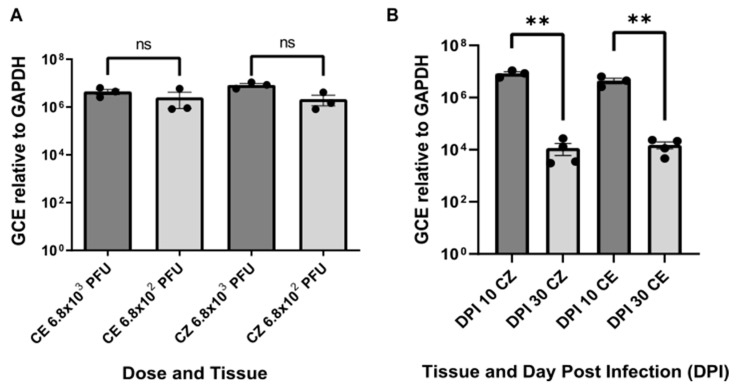
Genome Copy Equivalent (GCE) relative to GAPDH. (**A**) GCE of viral RNA in the cortex and cerebellum at 10 days post-inoculation (DPI) for two different inoculation doses. Viral copies per nanogram of total RNA (Y-axis) are shown for each tissue and dose (X-axis). Statistical significance was assessed using the Mann–Whitney U test, with “ns” indicating no significant differences (*p* > 0.05). (**B**) Comparison of ZIKV GCE in the cortex and cerebellum at 10 and 30 DPI. The number of equivalent viral copies (Y-axis) is plotted against the days post-inoculation (DPI) and the tissue type (CZ: cortex, CE: cerebellum). Statistical significance was determined using the Mann–Whitney U test, with ** indicating *p* < 0.01. The viral dose and DPI 10 experiments included three biological samples with three technical replicates each, while DPI 30 had four biological samples with three technical replicates.

**Figure 2 viruses-17-00405-f002:**
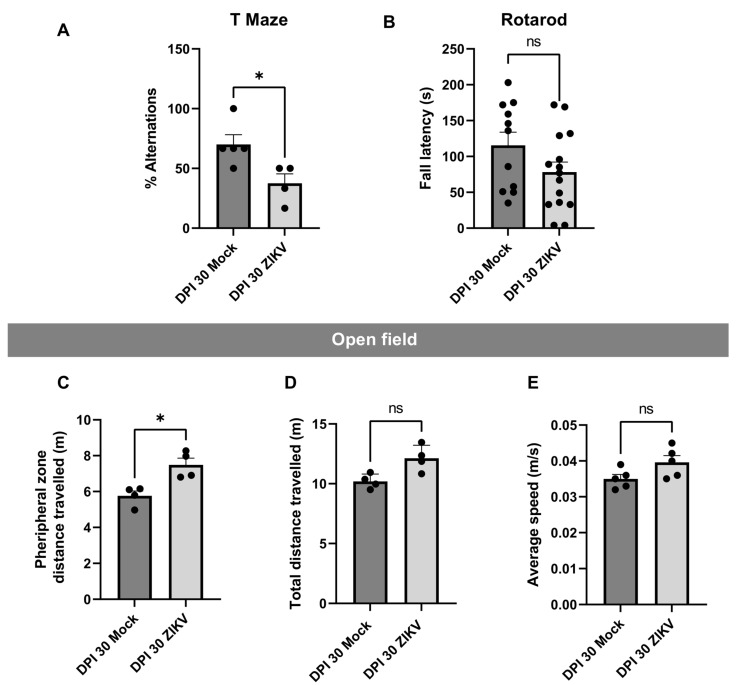
Behavioral Tests. (**A**) T-Maze: The percentage of alternations is shown for each evaluated group. Five biological samples were used. Each mouse performed an initial trial (T0) followed by six repetitions. (**B**) Rotarod: In seconds (s), the latency to fall is represented for each group and tissue. Five biological samples with three technical replicates each. (**C**) Open Field Test–Distance in Peripheral Zone: The distance traveled in meters (m) in the peripheral zone is shown. (**D**) Open Field Test–Total Distance: The total distance traveled during the test, in meters (m), is represented. (**E**) Open Field Test–Average Speed: The average speed in meters per second (m/s) of the individuals during the test is shown. Open Field: five biological samples with one technical replicate. In all cases, the Y-axis represents the evaluated attribute, and the X-axis represents the infection group and DPI. Statistical significance was determined using the Mann–Whitney U test, with dispersion shown as mean ± SEM. A single asterisk (*) indicates *p* < 0.05, and ns indicates *p* > 0.05. In all figures, the mock group is represented by the dark gray bar and the infected group by the light gray bar.

**Figure 3 viruses-17-00405-f003:**
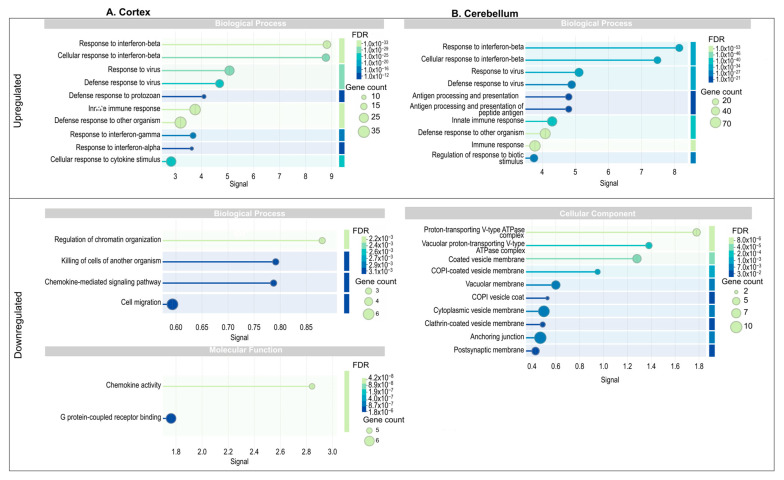
Functional enrichment analysis of protein–protein interaction network in cortex and cerebellum. Go terms associated with biological processes (BP), molecular functions (MF), and cellular components (CC) for upregulated and downregulated genes. The size of the circles indicates the number of genes associated with the term, while the color represents the FDR value, where darker colors indicate higher statistical significance. (**A**) Cortex; (**B**) Cerebellum.

**Figure 4 viruses-17-00405-f004:**
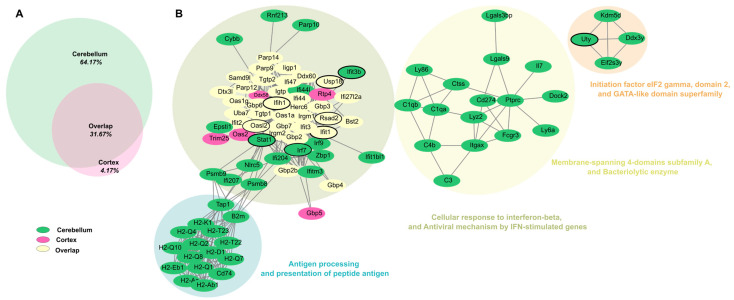
Merged network and distribution of upregulated DEGs in the cortex and cerebellum of zika virus-infected Mice. (**A**) Merged network of DEGs upregulated in cortex and cerebellum. Nodes are colored according to tissue of origin: green=cerebellum; fuchsia = cortex; and cream-yellow= genes shared between both tissues (overlap). Networks were grouped using the MCL clustering algorithm, representing functional groups. Hub genes are outlined in black, highlighting their key role within the networks. (**B**) Relationship illustrating the proportion of genes unique to each tissue and the shared genes (31.67%) between cortex and cerebellum.

**Figure 5 viruses-17-00405-f005:**
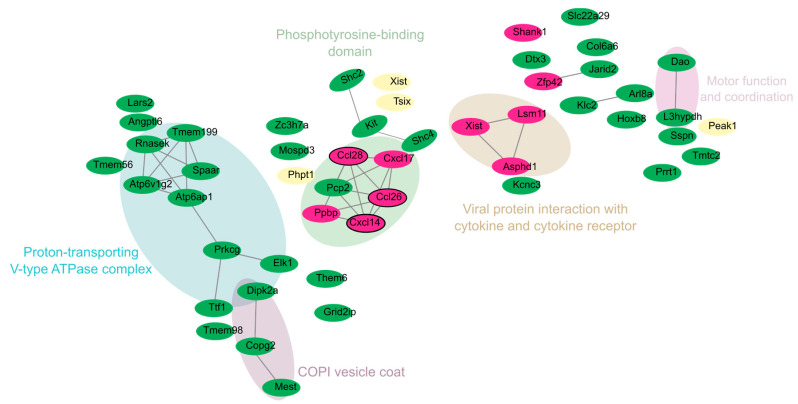
Merged network and distribution of downregulated DEGs in the cortex and cerebellum of zika virus-infected Mice. Nodes are colored according to tissue of origin: green = cerebellum; fuchsia = cortex; and cream-yellow = genes shared between both tissues (overlap). Networks were grouped using the MCL clustering algorithm, representing functional groups. Hub genes are outlined in black, highlighting their key role within the networks.

**Table 1 viruses-17-00405-t001:** Primers and Probes for Genome Detection and Quantification.

Target	Primer/Probe	RefSeq ID
ZIKV	F: CTGYGGGATCTCCTCTGTYTCAA	KJ776791.2
	R: ACGGGCAATCTCTGTGGASCTCT	
	P: FAM-ACGGTCGTTGTGGGATCTGTRAAA-BHQ-0	
GAPDH	F: AGGTCGGTGTGAACGGATTTG	NM_017008
	R: GGGGTCCGTTGATGGCAACA	

**Table 2 viruses-17-00405-t002:** Top 10 Hub Upregulated Genes and Their Functional Characteristics in the Cortex and Cerebellum of Zika Virus-Infected Mice.

Tissue	Gen	Uniprot Description	Uniprot Accession	Degree	Betweenness Centrality	Closeness Centrality
Cerebellum; Cortex	Ifit3	IFN-induced antiviral protein, which acts as an inhibitor of cellular, as well as viral processes, cell migration, proliferation, signaling, and viral replication.	Q64345	37	0.07	0.49
Cerebellum	Stat1	Signal transducer and transcription activator that mediates cellular responses to interferons (IFNs), cytokine KITLG/SCF, and other cytokines and other growth factors.	P42225	35	0.31	0.57
Cerebellum; Cortex	Ifit1	Interferon-induced antiviral RNA-binding protein that specifically binds single-stranded RNA bearing a 5′-triphosphate group (PPP-RNA), thereby acting as a sensor of viral single-stranded RNAs and inhibiting expression of viral messenger RNAs.	Q64282	32	0.15	0.53
Cerebellum; Cortex	Usp18	Interferon-induced ISG15 specific protease that plays a crucial role in maintaining a proper balance of ISG15-conjugated proteins in cells.	Q9WTV6	30	0.03	0.46
Cerebellum; Cortex	Ifih1	An innate immune receptor, which acts as a cytoplasmic sensor of viral nucleic acids and plays a major role in sensing viral infection and in the activation of a cascade of antiviral responses, including the induction of type I interferons and pro-inflammatory cytokines.	Q8R5F7	30	0.04	0.47
Cerebellum; Cortex	Ifi44	Interferon-induced protein 44; This protein aggregates to form microtubular structures; Belongs to the IFI44 family.	Q8BV66	29	0.02	0.46
Cerebellum; Cortex	Ifit2	IFN-induced antiviral protein, which inhibits expression of viral messenger RNAs lacking 2′-O-methylation of the 5′ cap.	Q64112	26	0.01	0.46
Cerebellum; Cortex	Rsad2	Interferon-inducible antiviral protein, which plays a major role in the antiviral state of the cell induced by type I and type II interferon.	Q8CBB9	23	0.04	0.46
Cerebellum; Cortex	Oasl2	Interferon-induced, dsRNA-activated antiviral enzyme, which plays a critical role in cellular innate antiviral response.	Q9Z2F2	23	0.02	0.44
Cerebellum	Irf7	Key transcriptional regulator of type I interferon (IFN)-dependent immune responses and plays a critical role in the innate immune response against DNA and RNA viruses. Regulates the transcription of type I IFN genes and IFN-stimulated genes (ISG) by binding to an interferon-stimulated response element (ISRE) in their promoters.	P70434	22	0.02	0.44

**Table 3 viruses-17-00405-t003:** Top 10 Hub Downregulated Genes and Their Functional Characteristics in the Cortex and Cerebellum of Zika Virus-Infected Mice.

Tissue	Gen	Uniprot Description	Uniprot Accession	Degree	Betweenness Centrality	Closeness Centrality
Cortex	Ccl28	C-C motif chemokine 28; Chemotactic for resting CD4, CD8 T-cells, and eosinophils. Binds to CCR10 and induces calcium mobilization in a dose-dependent manner.	Q9JIL2	4	0.06	1.00
Cortex	Ccl26	Chemokine (C-C motif) ligand 26	F8VQM2	4	0.06	1.00
Cortex	Cxcl14	Chemotactic for CESS B-cells and THP-1 monocytes, but not T-cells.	Q9WUQ5	4	0.06	1.00
Cerebellum	Atp6ap1	Guides the V-type ATPase into specialized subcellular compartments, such as neuroendocrine regulated secretory vesicles or the ruffled border of the osteoclast, thereby regulating its activity. Involved in membrane trafficking and Ca^2+^-dependent membrane fusion.	Q9R1Q9	4	0.57	0.70
Cerebellum	Atp6v1g2	V-type proton ATPase subunit G 2; Catalytic subunit of the peripheral V1 complex of vacuolar ATPase (V-ATPase). V-ATPase is responsible for acidifying a variety of intracellular compartments in eukaryotic cells.	Q9WTT4	4	0.06	0.58
Cerebellum	Rnasek	Regulates the activity of vacuolar (H+)-ATPase (V-ATPase), which is responsible for acidifying and maintaining the pH of intracellular compartments. Required at an early stage of receptor-mediated endocytosis.	Q8K3C0	4	0.06	0.58
Cerebellum	Tmem199	Accessory component of the proton-transporting vacuolar (V)-ATPase protein pump involved in intracellular iron homeostasis. In aerobic conditions, required for intracellular iron homeostasis, thus triggering the activity of Fe2+ prolyl hydroxylase (PHD) enzymes and leading to HIF1A hydroxylation and subsequent proteasomal degradation.	Q5SYH2	4	0.06	0.58
Cerebellum	Prkcg	Calcium-activated, phospholipid- and diacylglycerol (DAG)-dependent serine/threonine protein kinase that plays diverse roles in neuronal cells and eye tissues, such as regulation of the neuronal receptors GRIA4/GLUR4 and GRIN1/NMDAR1.	P63318	3	0.52	0.58
Cerebellum	Spaar	Negatively regulates mTORC1 activation by inhibiting recruitment of mTORC1 to lysosomes upon stimulation with amino acids: acts by promoting the formation of a tightly bound supercomplex composed of the lysosomal V-ATPase, Ragulator, and Rag GTPases, preventing recruitment of mTORC1	A0A1B0GSZ0	3	0.00	0.44
Cortex	Shank1	Seems to be an adapter protein in the postsynaptic density (PSD) of excitatory synapses that interconnects receptors of the postsynaptic membrane, including NMDA-type and metabotropic glutamate receptors, and the actin-based cytoskeleton. Plays a role in the structural and functional organization of the dendritic spine and synaptic junction.	D3YZU1	1	0.00	1.00

## Data Availability

The data presented in this study are openly available in BioProject, NIH. https://www.ncbi.nlm.nih.gov/bioproject/PRJNA1207202 accessed on 10 march 2025.

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
