# Peer review of "Mild Zika Virus Infection in Mice Without Motor Impairments Induces Working Memory Deficits, Anxiety-like Behaviors, and Dysregulation of Immunity and Synaptic Vesicle Pathways"

_viruses, 2025, doi:10.3390/v17030405_

Round 1
Reviewer 1 Report
Comments and Suggestions for Authors
Title. The title implies that immune activation and synaptic dysfunction contribute to cognitive impairments. However, the data indicates that these and many other DEGs are associative, but not causational. Immune activation is certainly expected in ZIKV infection. One could justify that immune activation and synaptic dysfunction are associated with the impairments but stating that they underlie the impairments is not accurate, in my opinion. The title ought to be changed to represent the outcome of the study.
Abstract
Line 22. “effectively replicate” implies that nearly every aspect of motor and cognitive deficits are the same as in humans. One might state that the murine infection models have some important aspects of motor and cognitive deficits in humans.
Results
line 304. This sentence seems to contradict itself, "each infected animal exhibited these signs with varying degrees of severity, ranging from completely healthy phenotypes to more apparent, yet non-severe, disease states." If some animals had healthy phenotypes, should it state that each animal had at least some disease phenotypes. Please clarity.
Figure 1.
The acronym for genome quantification equivalent is stated as GCE. Is this a typo? Should it e GQE?
Figure 2B.
Show the unit of sec for the Fall Latency. The text (line 336) states that the mean latency for the sham group was 90 sec, but it was 78 sec for the ZIKV-infected group. The text and the figures do not agree. Please clarify.
Because this figure shows important readouts for validation of the mouse model particularly with Fig. 1A and 1C, I would really like to see the individual data points superimposed on the bar graphs as done in Figure 1. I realize that a p value < 0.05 could mean a p values ranging below 0.05. Seeing the actual distribution and the number of data points would give me more confidence that these are true biological effects. The best scenario would be to see these data in a replicate experiment. These data could be the most valuable contribution of the paper, but the reader needs to be convinced.
Figure 2C. The text (line 339) of 51.1 meters does not agree with the figure. Should it read 5.1?
line 514. Citations are given to support the idea that down regulation of cellular metabolism and synaptic activity can possibly disrupt neuronal communications and plasticity, but describing some published data specifically supporting this would be helpful, in addition to the cited references provided. From the list of cited references, I do not see specific evidence that down regulation of cellular metabolism and synaptic function might cause disruption neuronal communications and plasticity.
line 517. Remove the period after plasticity. Also, the reference should be [50-54].
line 525. insert space after [55]
line 356. insert space "tissuesthe"
Discussion.
1st and 2nd paragraphs. Please discuss any other ZIKV-mouse models for behavioral neurodevelopmental effects. How does your model compare to them?
line 516. The authors suggest potential disruption in neuronal communication and plasticity based on the down regulation of Shank1 and Shank2. Shank1 is a down regulated DEG, but I cannot find any data in the paper or in Table S1 showing a down regulation of Shanks2. Without having identified other network genes that regulate neuronal function or plasticity, and validation of down regulation of both Shank1 and Shank2 by quantitative RT-PCR, I believe the title and the Shank1 and Shank2 discussion is an overstatement. The authors might consider running their DEGs in SynGo consortium to find any more synaptic DEGs. If the hypothesis of disruption of neuronal communications and plasticity cannot be strengthened, the reference to neuronal dysfunction needs to be removed or minimized in the title, abstract, and the Discussion conclusion.
The authors focus on downregulated DEGs of cellular metabolism and synaptic function that may disrupt neuronal function. Certain upregulated DEGs could conceivably affect these functions too. Were any of these identified?
Any limitations or caveats of this report should be discussed, like for example, changes in mRNA expression do not necessarily indicate that the protein levels are altered. Also, DEG identification by bioinformatics is best validated by measuring individual RNAs with northern blots, RT-qPCR or etc.
Author Response
Comment 1: Title. The title implies that immune activation and synaptic dysfunction contribute to cognitive impairments. However, the data indicates that these and many other DEGs are associative, but not causational. Immune activation is certainly expected in ZIKV infection. One could justify that immune activation and synaptic dysfunction are associated with the impairments but stating that they underlie the impairments is not accurate, in my opinion. The title ought to be changed to represent the outcome of the study.
Response 1: Thank you for your observation. We have revised the title to more accurately reflect the findings of the study. The new title has been changed to [Mild Zika Virus Infection in Mice Induce Working Memory Deficits, Anxiety-like Behaviours and dysregulation of genes related to immune response and Synaptic Vesicle Activity]. This change has been implemented on page 1, line 2-4, and is highlighted in yellow in the manuscript.
Comment 2: Abstract
Line 22. “effectively replicate” implies that nearly every aspect of motor and cognitive deficits are the same as in humans. One might state that the murine infection models have some important aspects of motor and cognitive deficits in humans.
Response 2: Response 2: Thank you for pointing this out. The phrase "effectively replicate" has been replaced to [can be used to mimic part of the spectrum of] to more accurately reflect the relationship between the murine model and the motor and cognitive deficits observed in humans. This change has been made on page 1, line 21-22.
Additionally, we removed the references to synaptic function that were previously included in the abstract.
Comment 3: line 304. This sentence seems to contradict itself, "each infected animal exhibited these signs with varying degrees of severity, ranging from completely healthy phenotypes to more apparent, yet non-severe, disease states." If some animals had healthy phenotypes, should it state that each animal had at least some disease phenotypes. Please clarity.
Response 3: Thank you for your comment. We recognize that the wording may have been ambiguous and have revised it for clarity. For accuracy, the text now reads as follows:
[The group of infected animals showed individuals with varying degrees of clinical signs, ranging from no noticeable symptoms to mild manifestations of the disease.] This change has been made on page 8, line 357-359, and is highlighted in yellow in the manuscript.
Comment 4: Figure 1.
The acronym for genome quantification equivalent is stated as GCE. Is this a typo? Should it be GQE?
Response 4: This was a typographical error in the definition of the acronym. The acronym GCE was correct, but the definition was incorrect. It has now been corrected in the text as [GCE – Genome Copy Equivalents], instead of Genome Quantification Equivalent. This correction has been made on page 8, line 375, and is highlighted in yellow.
Comment 5: Figure 2B.
Show the unit of sec for the Fall Latency. The text (line 336) states that the mean latency for the sham group was 90 sec, but it was 78 sec for the ZIKV-infected group. The text and the figures do not agree. Please clarify.
Response 5: Thank you for your observation. The unit of seconds (s) has been added to the graph, as seen on page 10, line 402, Figure 2B. The previously reported value of 90 seconds for the control group was a typographical error; it has now been corrected and replaced with [115] seconds. This change has been implemented on page 9, line 395, and is highlighted in yellow.
Comment 6: Because this figure shows important readouts for validation of the mouse model particularly with Fig. 1A and 1C, I would really like to see the individual data points superimposed on the bar graphs as done in Figure 1. I realize that a p value < 0.05 could mean a p values ranging below 0.05. Seeing the actual distribution and the number of data points would give me more confidence that these are true biological effects. The best scenario would be to see these data in a replicate experiment. These data could be the most valuable contribution of the paper, but the reader needs to be convinced.
Response 6: Thank you for your observation. Individual data points have been added for each data set in Figure 2. These changes have been implemented on page 10, line 402, Figure 2B.
Comment 7: Figure 2C. The text (line 339) of 51.1 meters does not agree with the figure. Should it read 5.1?
Response 7: Thank you for noticing this. There was a typographical error, which has now been corrected. The incorrect value 51.1 m has been replaced with [5.8] m in the text. This change has been made on page 9, line 398.
Comment 8: line 514. Citations are given to support the idea that down regulation of cellular metabolism and synaptic activity can possibly disrupt neuronal communications and plasticity, but describing some published data specifically supporting this would be helpful, in addition to the cited references provided. From the list of cited references, I do not see specific evidence that down regulation of cellular metabolism and synaptic function might cause disruption neuronal communications and plasticity.
Respuesta 8: Thank you for your observation. To strengthen the argument and provide more specific evidence, additional citations have been included with published data supporting the relationship between the dysregulation of cellular metabolism, synaptic activity, and its impact on neuronal communication and plasticity.
Additionally, the paragraph has been incorporated into the manuscript to reinforce this information. This change has been made on page 18, line 664-677, and is highlighted in yellow.
Comment 9: line 517. Remove the period after plasticity. Also, the reference should be [50-54].
Response 9: The wording has been changed. Citation was updated due to the inclusion of new references. Changes made on page 18, line 664.
Comment 10: line 525. insert space after [55]
Response 10: Thank you for your observation. See Page 18, line 686. This citation is now [64].
Comment 11: line 356. insert space "tissuesthe".
Response 11: Thank you for pointing this out. The typographical error has been corrected; the text now reads [tissues the]. Page 11, line 420.
Comment 12: Discussion.
1st and 2nd paragraphs. Please discuss any other ZIKV-mouse models for behavioral neurodevelopmental effects. How does your model compare to them?
Response 12: Two texts were added to the discussion regarding this topic, see page 17, line 617-628 and line 638-649.
Comment 13. line 516. The authors suggest potential disruption in neuronal communication and plasticity based on the down regulation of Shank1 and Shank2. Shank1 is a down regulated DEG, but I cannot find any data in the paper or in Table S1 showing a down regulation of Shanks2. Without having identified other network genes that regulate neuronal function or plasticity, and validation of down regulation of both Shank1 and Shank2 by quantitative RT-PCR, I believe the title and the Shank1 and Shank2 discussion is an overstatement. The authors might consider running their DEGs in SynGo consortium to find any more synaptic DEGs. If the hypothesis of disruption of neuronal communications and plasticity cannot be strengthened, the reference to neuronal dysfunction needs to be removed or minimized in the title, abstract, and the Discussion conclusion.
Response 13: Thank you for pointing this out. We acknowledge that including the gene Shank2 was overestimated. As the reviewer correctly noted, Shank2 is not among the downregulated genes in our dataset. This arose from the KEGG-based enrichment analysis used to generate the graphical network. However, we confirm that Shank2 is not part of our list of differentially expressed genes, and thus, it has been removed from the analysis.
To soften the statement, the words [may] and [suggest] were added in lines 601 and 604, page 17 within the results section, and a paragraph was added to the discussion on this topic. See on page 18, line 729-735. This change, highlighted in yellow.
Comment 14: The authors focus on downregulated DEGs of cellular metabolism and synaptic function that may disrupt neuronal function. Certain upregulated DEGs could conceivably affect these functions too. Were any of these identified?
Response14 Yes, all the upregulated genes are involved in immune response. However, certain immunological mechanisms, such as neuroinflammation, can negatively impact neurological function. To clarify this point, we have added two paragraphs to the manuscript: Page 18, Line 664-677, and line 693-698.
Comment 15: Any limitations or caveats of this report should be discussed, like for example, changes in mRNA expression do not necessarily indicate that the protein levels are altered. Also, DEG identification by bioinformatics is best validated by measuring individual RNAs with northern blots, RT-qPCR or etc.
Response 15: Thank you for your observation. A paragraph on limitations and perspectives has been added to the manuscript. Changes have been incorporated on page 19, line 754-765.
Reviewer 2 Report
Comments and Suggestions for Authors
In this manuscript, the authors established a model system in mice that simulates the mild congenital ZIKV infection human, to study the behavioral changes and molecular mechanism following subtle ZIKV infection. The overall finding is interesting and provided insights for further investigation. With some minor revisions, this manuscript can be considered to be published in Viruses.
Key concerns to be addressed are listed below:
- Among the 5 animal that the authors picked out that appeared to be healthy, have they taken the measurements of their brain? Did the animals show any sign of microcephaly?
- In Figure 1, please adjust the y-axis to start with 0. Truncated y-axis can be misleading when presenting the data. Log scale is an option as well.
- In Figure 3, the location of the label "Downregulated" is a bit confusing. Does it apply to both 3A and 3B? Does it apply to "Molecular function" section? Please revise so it is easier for the audience to understand.
- Please specify the sample number and # of replicates in all the Figures that applies.
- Line 356, typo "tissuesthe".
- None of the upregulated or downregulated genes are validated in protein level. Will the authors be able to maybe validate some of the top hits? Especially the ones that are specified to each region?
- In the discussion section, I recommend the the author discuss if any previous work support or disagree with the genes identified.
Author Response
Comment 1: Among the 5 animal that the authors picked out that appeared to be healthy, have they taken the measurements of their brain? Did the animals show any sign of microcephaly?
Response 1: Thank you for the observation. In this study, brain measurements were not performed. However, no apparent changes were observed in the size of the animals, signs of microphthalmia, or motor alterations such as paralysis and tremors. On the other hand, our research group, using this same viral strain at higher doses and inoculating via the intraperitoneal route, was able to develop an infection model with morphological changes in size, weight, and ocular impairments (Rengifo et al., 2023; Rivera et al., 2019) see page 17, lines 617-628.
Additionally, Paul et al. (2018) also reported a decrease in brain size in a mild prenatal infection model, in which the offspring did not present severe congenital abnormalities, had no detectable viral presence in the brain, and displayed body weights comparable to controls. This additional information has been included in the introduction section see on page 2, line 83-85, and has been highlighted in yellow in the manuscript.
Comment 2: In Figure 1, please adjust the y-axis to start with 0. Truncated y-axis can be misleading when presenting the data. Log scale is an option as well.
Response 2: Thank you for the observation. The y-axis in Figures 1A and 1B has been adjusted to a logarithmic scale (base 10), starting at 100. This adjustment allows for a better comparison between both graphs, eliminating the bias caused by the scale difference previously present in Figure 1A. The modification has been incorporated on page 8, line 373.
Comment 3: In Figure 3, the location of the label "Downregulated" is a bit confusing. Does it apply to both 3A and 3B? Does it apply to "Molecular function" section? Please revise so it is easier for the audience to understand.
Response 3: Thank you for the observation. The figure has been adjusted to improve clarity and facilitate the interpretation of the "Downregulated" label. Each section has been outlined in black to more precisely indicate the corresponding images. This adjustment has been incorporated on page 12, line 464.
Comment 4: Please specify the sample number and # of replicates in all the Figures that applies.
Response 4: Thank you for your observation. The description of Figure 1 has been updated to read:
"[The viral dose and DPI 10 experiments included three biological samples with three technical replicates each, while DPI 30 had four biological samples with three technical replicates.]" Page 8, line 382-384, Figure 1.
In Figure 2, individual data points have been added. The changes have been incorporated on line 402.
The figure descriptions now specify the number of biological samples and technical replicates for each experiment as follows:
"[Five biological samples were used. Each mouse performed an initial trial (T0) followed by six repetitions.]" Page 10, line 404-407.
"[Five biological samples with three technical replicates each.]" Page 10, line 406.
"[Open field: five biological samples with one technical replicate.]" Page 10, line 410-411.
These modifications have been highlighted in yellow in the manuscript.
Comment 5: Line 356, typo "tissuesthe".
Response 5: Thank you for pointing this out. The typographical error has been corrected; the text now reads [tissues the]. Page 11, line 420.
Comment 6: None of the upregulated or downregulated genes are validated in protein level. Will the authors be able to maybe validate some of the top hits? Especially the ones that are specified to each region?
Response 6: Thank you for your observation. In this study, protein-level validation of the identified genes was not performed due to the high number of differentially expressed genes found. It is important to emphasize that the primary objective of this study was to identify gene networks with potential as targets for future research. The findings presented here are intended to serve as a foundation for subsequent studies that will validate the protein expression of these genes.
Additionally, we have clarified this point in the manuscript. See page 19-20, line 754-766 of the manuscript.
Comment 7: In the discussion section, I recommend the author discuss if any previous work support or disagree with the genes identified.
Response 7: Three paragraphs were added. Page 18 line 655-662;page 19, line 708-714; page 19 line 716-727.
Round 2
Reviewer 1 Report
Comments and Suggestions for Authors
The authors have satisfactorally answered all of my concerns.